# Integrating Soil pH, Clay, and Neutralizing Value of Lime into a New Lime Requirement Model for Acidic Soils in China

**Dandan Han [1,2], Saiqi Zeng [3], Xi Zhang [1], Jumei Li [1] and Yibing Ma [4,*]**

[1]  Institute of Agricultural Resources and Regional Planning, Chinese Academy of Agricultural Sciences, Beijing 100081, China; handandan159753@163.com (D.H.); sunnyx321@163.com (X.Z.); lijumei@caas.cn (J.L.)
[2]  College of Resources and Environmental Sciences, China Agricultural University, Beijing 100193, China
[3]  Institute of Soil and Water Resources and Environmental Science, College of Environmental and Resource Sciences, Zhejiang University, Hangzhou 310058, China; sqzeng@zju.edu.cn
[4]  Macau Environmental Research Institute, Macau University of Science and Technology, Macau 999078, China
[*]  Correspondence: ybma@must.edu.mo; Fax: +853-28833280

**Abstract:** Modelling the lime requirement (LR) is a fast and efficient way to determine the amount of lime required to obtain a pH that can overcome the adverse effects caused by soil acidification. This study aimed to model the LR based on the properties of soil and lime. A total of 17 acidic soils and 39 lime samples underwent soil–lime incubation in the laboratory. The predictive equations for the LR (t ha$^{-1}$) were modelled using $\Delta$pH (the difference between the target pH and initial pH), the neutralizing value (NV, mmol kg$^{-1}$) of lime, soil pH, soil clay content (%), soil bulk density (BD, g cm$^{-3}$), and the depth of soil (h, cm) as the factors in an exponential equation. The generic predictive equation, $LR = \Delta pH \times e^{-3.88-0.069 \times NV+0.51 \times pH+0.025 \times Clay} \times BD \times h$, was validated as the most reliable model under field conditions. Simplified predictive equations for different soil textures when limed with quicklime and limestone are also provided. Furthermore, the LR proportions provided by hydrated lime, quicklime, limestone, and dolomite in commercially available lime can be expressed as 0.58:0.64:0.97:1.00. This study provides a novel and robust model for predicting the amount of lime product containing components with different neutralizing abilities that are required to neutralize soils with a wide range of properties. It is of great significance to agronomic activities and soil remediation projects.

**Keywords:** buffer capacity; exponential; liming; pedotransfer function; soil acidification





## 1. Introduction

Liming is a traditional and effective way to mitigate the adverse impacts caused by soil acidification, such as the loss of nutrient elements ($NO_3^-$, $SO_4^{2-}$, etc.) [1,2] and activation of toxic elements (Al, Mn, Cd, etc.) [3,4]. It can lead to enhance soil quality, crop productivity, and food safety [5–7]. The lime requirement (LR) is the amount of lime required to increase the pH of an acidic soil to a desired value that is suitable for plant growth [8]. It is critical when attempting to improve adverse crop growth conditions in acidic soils [9].

Modelling the LR is a fast and simple way to determine the amount of lime that needs to be applied to soil to attain the target pH. Previous studies have revealed that the LR can be predicted from soil properties, such as soil pH, organic matter (OM), and potential acidity [10,11], using pedotransfer functions (PTFs), which are easily, routinely, or inexpensively measured soil properties [12]. Several linear and nonlinear predictive equations for the LR have been modelled using PTFs [10,11,13].

Soil pH buffer capacity (pHBC) is an intrinsic soil property that quantifies the ability of soil to resist pH change upon the addition of acidity or alkalinity [14]. Modelling the LR via pHBC is a widely accepted principle and commonly used modelling method [15,16]. To facilitate modelling, pHBC is usually described in terms of the lime buffer capacity (LBC), which is the weight of lime required to change the soil pH by one unit (mg kg$^{-1}$ pH$^{-1}$) [15,17,18]. The

LBC can be estimated via PTFs that have widely available data sets or are relatively easily measured soil properties [19,20]. Previous studies have shown that clay is an important soil property for pHBC estimation [21,22] and LR prediction [13,21–24]. Soil initial pH and target pH, which are the indicators of the present soil acidity and the desired acidity, respectively, are also important factors in LR prediction [25,26]. The neutralizing ability of lime is also a key factor affecting the LR [24,27–31].

The current LR predictive models have usually been established for specific purposes, such as for tropical soils [32], Brazil [33], permanent grassland, and arable crops [34]. The LR is usually expressed as the amount of pure lime, such as pure $CaCO_3$ [35,36] and pure $Ca(OH)_2$ [15,17,37], although the LR can be converted to another unit using the relative neutralizing value, effective calcium carbonate equivalent [30], or a calibration model [10]. Significant acidification has been found in major Chinese croplands [38–40], and a vast array of red soil needs liming, especially in Hunan. However, there is no model for predicting the required amount of lime product that needs to be supplied to soils in China when the products contain components that have a wide range of neutralizing abilities and/or the soils have a wide range of different properties. This study aimed to establish a model that integrated soil and lime properties to predict the required amount of lime product that needs to be applied to soils in China to attain a desired soil pH when the products contain components that have a wide range of neutralizing abilities and/or the soils have a wide range of soil properties.

## 2. Materials and Methods

### 2.1. Collection and Characterization of Soil and Lime Samples

Seventeen representative acidic soils were collected from the topsoil layer (0~20 cm) in south China, including six soils in Hunan and one or two soils in every other province where the soil pH is low. All soil samples were provided by local soil management department. A further 39 samples of different types of lime, including quicklime, hydrated lime, limestone, and dolomite, were also collected from factories and markets across China.

The soil samples were air dried, ground, and sieved using a 2 mm sieve (10 mesh). The soil pH was determined in a deionized water suspension with a 1:5 soil-to-water ratio, the OM was determined using the potassium dichromate oxidation outer heating method [41], and the particle size distribution was determined using the pipette method [42].

The lime samples were ground and sieved through a 0.15 mm mesh sieve (100 mesh), and then the Ca and Mg in the lime samples were determined using an ICP mass spectrometer after they had been digested with $HNO_3$, HF, and $HClO_4$ in a pressure digester at 120 °C.

Table S1 (see Supplementary Materials for Table S1) shows that the soil and lime samples had a broad range of properties. The soil pH ranged between 4.39 and 6.19 (mean = 5.05), and the clay content was between 4.40% and 60.40% (mean = 38.05%). The average content of calcium plus magnesium contents ([Ca+Mg]) in quicklime, hydrated lime, limestone, and dolomite was 64.14%, 55.52%, 42.41%, and 38.19%, respectively. Detailed information on the soil and lime samples is shown in Tables S2 and S3 (see Supplementary Materials for Tables S2 and S3).

### 2.2. Neutralizing Value of Lime

The NV of lime was determined using the acid dissolution and back titration method with modifications [43]. The experimental procedure was as follows: 0.200 g of lime and 10.00 mL of 1 mol $L^{-1}$ HCl were added to a beaker. The beaker was oscillated for 12 h, and then 10.00 mL of deionized water and two drops of phenolphthalein were added. Finally, the solution in the beaker was titrated with 1 mol $L^{-1}$ NaOH. There were double replications for each lime sample. The control (CK) treatment was no added lime. The NV

was expressed as the amount of H$^+$ neutralized by one kilogram of lime (mmol kg$^{-1}$) and was calculated using the following equation:

$$NV = \frac{(V_{CK} - V) \times C_{NaOH}}{W} \tag{1}$$

where V$_{CK}$ (mL) is the volume of NaOH consumed in the CK treatment, *V* (mL) is the volume of NaOH consumed in the lime treatment, C$_{NaOH}$ is the NaOH concentration, and W (g) is the weight of the lime.

### 2.3. Soil-Lime Incubation

There were two soil–lime incubation processes. The first was used to determine the extent to which the neutralizing ability of a lime product influenced the LR. Two soil samples (S1 and S2) with low soil pHs that required large amounts of lime were incubated with the 39 lime samples (L1~L39).

The second process was used to explore the influence of soil properties on the LR. Seventeen soil samples (S1~S17) were incubated with four different types of lime (quicklime, hydrated lime, limestone, and dolomite, L1~L4), which had a range of neutralizing abilities and were from different sources. Each soil was incubated with five incremental rates of lime. The rates of lime were based on the lime calculator ([http://www.aglime.org.uk/lime_calculator.php](http://www.aglime.org.uk/lime_calculator.php)) (accessed on 10 May 2023), which uses soil pH, soil texture, lime source, and NV to calculate the rates. It should be noted that the maximum rate of lime was 0.75%, and the soil–lime incubation process was conducted in a constant temperature incubator (25 °C). The experimental procedure was as follows: 10.00 g of soil and different rates of lime were placed in a centrifuge tube and mixed thoroughly. Then, the soil and lime were moistened to 70% of their field capacity with distilled water, and the soil moisture level was kept constant by adding appropriate distilled water at regular intervals based on the gravimetric method. After 60 days, the soil in each treatment was air dried and sieved (2 mm). Subsequently, 5.00 g of soil sample was placed in a centrifuge tube, and the soil pH was determined in deionized water (the soil-to-solution ratio was 1:5). The CK treatment was incubating the soil without lime. There were three replications in each treatment.

### 2.4. Modelling

To convert the results to the LR unit, it was assumed that the soil bulk density was 1.25 g cm$^{-3}$, and the depth of lime-neutralized soil was 12 cm, which meant that the weight of the surface soil was 1500 t ha$^{-1}$. In this present study, the exponential method, the LBC-exponential method, and 1/LBC-exponential method were used for LR modelling (Table 1). It should be noted that the actual LRs were the amount of lime added to the soil, the corresponding target pHs were the soil pHs after incubation, the initial pHs were the soil pHs before incubation, the ΔpH was calculated as the target pH minus the initial pH, the LBC was the slope of the linear relationship for LR vs. pH, and 1/LBC was the slope of the linear relationship for pH vs. LR. The parameters in each equation were optimized by minimizing the residual sum of squares between the predicted LR and actual LR using the least squares method. The following data sets were eliminated before modelling: (1) target pHs over 7.50; (2) ΔpHs less than 0.30; (3) ΔpHs that remained constant with the increasing addition of lime.

**Table 1.** Methods used to model LR.

| Method | Equation |
| --- | --- |
| Exponential method | $LR = e^{A + a \times F_1 + b \times F_2 + ... + x \times F_x} \times BD \times h$ |
| LBC-exponential method | $LR = \Delta pH \times e^{A + a \times F_1 + b \times F_2 + ... + x \times F_x} \times BD \times h$ |
| 1/LBC-exponential method | $LR = \Delta pH / e^{A + a \times F_1 + b \times F_2 + ... + x \times F_x} \times BD \times h$ |

LR is the lime requirement (t ha$^{-1}$), F$_1$, F$_2$, and F$_x$ are the soil and lime properties, BD is the soil bulk density (1.25 g cm$^{-3}$), and h is the depth of soil (12 cm).

### 2.5. Model Validation

The LR models were validated with field studies. Relevant published articles were searched for in the China National Knowledge Infrastructure (https://www.cnki.net/) (accessed on 10 May 2023) and the National Agricultural Information System (https://www.nais.net.cn/) (accessed on 10 May 2023) databases using the following keywords: lime, liming, and acid soil. A total of ten articles and 24 data sets were selected, and the detailed information is presented in Table S4 (see Supplementary Materials for Table S4). Prior to validation, the equation $pH_{1:5W} = 0.14 + 0.99 \times pH_{1:2.5W}$ was used to convert soil pH measured at a solution ratio of 1:2.5 $H_2O$ to 1:5 $H_2O$ [44].

### 2.6. Statistical Analysis

Linear correlations and regression analyses were performed using Excel 2016 (Microsoft, Redmond, WA, USA), and the figures were plotted via OriginPro 2018 (OriginLab, Northampton, MA, USA). The root mean squared error (RMSE), mean error (ME), mean absolute error (MAE), and mean absolute percentage error (MAPE) were calculated as follows:

$$RMSE = \sqrt{\frac{1}{n}\sum_{i=1}^{n}(\hat{x}_i - x_i)^2} \tag{2}$$

$$ME = \frac{1}{n}\sum_{i=1}^{n}(\hat{x}_i - x_i) \tag{3}$$

$$MAE = \frac{1}{n}\sum_{i=1}^{n}|\hat{x}_i - x_i| \tag{4}$$

$$MAPE = \frac{1}{n}\sum_{i=1}^{n}\left|\frac{\hat{x}_i - x_i}{x_i}\right| \tag{5}$$

where $x_i$, $\hat{x}_i$, and $n$ are the $i$th actual LR, the predicted LR, and the total number of samples, respectively.

## 3. Results

### 3.1. Relationship between Neutralizing Value and [Ca+Mg] in Lime

Figure 1 shows that there was a significant positive relationship between the NV and [Ca+Mg] in lime. This result implies that the NV of lime can be estimated by [Ca+Mg].

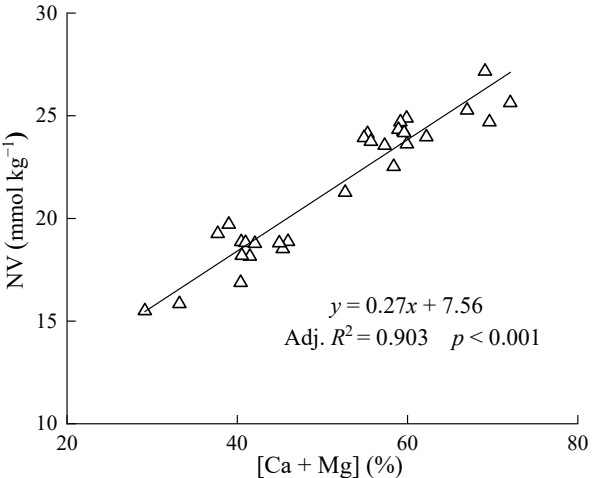

**Figure 1.** Relationship between NV and the [Ca+Mg] in lime.

Table S5 (see Supplementary Material for Table S5) shows that the mean NVs of quicklime, hydrated lime, limestone, and dolomite were 24.68, 23.58, 18.34, and 18.20 mmol kg$^{-1}$, respectively. This result indicates that the order for the average neutralizing ability of

the tested commercially available lime products was quicklime > hydrated lime > limestone > dolomite. The average neutralizing ability proportions for the quicklime, hydrated lime, limestone, and dolomite were 1.00:0.96:0.74:0.74, indicating that quicklime had the maximum neutralizing ability, and the neutralizing ability of limestone is similar to that of dolomite.

### 3.2. Relationships between LBC or 1/LBC and Soil pH, Clay Content, the NV of Lime

Figure 2 shows that there were negative correlations between the LBC and NV of lime but positive correlations between 1/LBC and the NV of lime. The slopes of the linear relationships for soil S1 and soil S2 were different, which indicated that the LBC and 1/LBC were affected by both the NV of lime and soil properties.

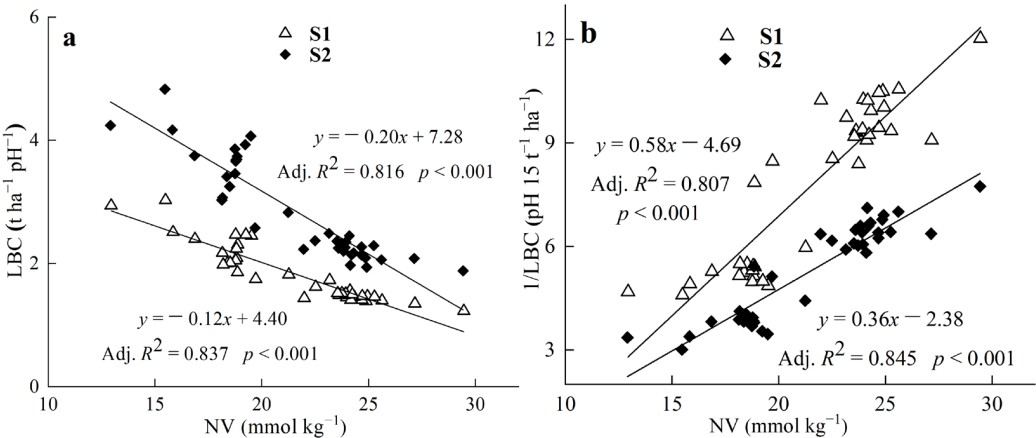

**Figure 2.** Relationships between LBC (**a**) or 1/LBC (**b**) and NV of lime.

Table 2 shows that the LBC of the different types of lime was in the order of dolomite > limestone > hydrated lime > quicklime, implying that the amount of dolomite required to change the soil by one pH unit was the greatest and the amount of quicklime required was lowest. The average LBC proportions for quicklime, hydrated lime, limestone, and dolomite were 0.58:0.64:0.97:1.00, implying that the proportional amounts of hydrated lime, quicklime, limestone, and dolomite in commercial lime required for soil to change by one unit pH was 0.58:0.64:0.97:1.00.

**Table 2.** Descriptive statistics of LBCs ($t\ ha^{-1}\ pH^{-1}$) of the different types of lime in soil S1 and S2.

| Statistical Indicator | Quicklime | | Hydrated Lime | | Limestone | | Dolomite | |
|---|---|---|---|---|---|---|---|---|
| | S1 | S2 | S1 | S2 | S1 | S2 | S1 | S2 |
| Mean | 2.19 | 1.47 | 2.36 | 1.68 | 3.50 | 2.58 | 3.72 | 2.58 |
| Standard deviation | 0.04 | 0.02 | 0.10 | 0.08 | 0.15 | 0.11 | 0.23 | 0.13 |
| Median | 2.21 | 1.45 | 2.31 | 1.61 | 3.53 | 2.63 | 3.86 | 2.72 |
| Minimum | 1.98 | 1.39 | 1.96 | 1.46 | 2.44 | 1.86 | 2.59 | 1.75 |
| Maximum | 2.36 | 1.60 | 3.19 | 2.35 | 4.25 | 3.15 | 4.84 | 3.02 |
| Relative value | 0.59 | 0.57 | 0.63 | 0.65 | 0.94 | 1.00 | 1.00 | 1.00 |

The equations were evaluated using the coefficient of determination ($R^2$), and the results showed that exponential models accurately estimated the LBC using the NV of lime, soil pH, and clay content (Table 3). Therefore, it is feasible to estimate the LBC from the NV of lime, soil pH, and clay content using the exponential models.

**Table 3.** Equations for LBC (t ha$^{-1}$ pH$^{-1}$) and 1/LBC (pH 15 t$^{-1}$ ha$^{-1}$) described using the NV (mmol kg$^{-1}$) of lime, pH, and clay content (%).

| Equation | Predictive Equation | $R^2$ | $n$ |
|:---:|:---:|:---:|:---:|
| (6) | $\mathrm{LBC} = e^{-0.41-0.089\times NV+0.56\times pH+0.014\times Clay}$ | 0.75 | 122 |
| (7) | $1/\mathrm{LBC} = e^{0.96+0.10\times NV-0.41\times pH-0.011\times Clay}$ | 0.82 | 122 |

### 3.3. Modelling Based on Soil-Lime Incubation

Table 4 shows the generic predictive equations for LR modelled using the exponential methods. With the NV of lime, ΔpH, pH, and clay content, the generic predictive equations modelled using three methods can accurately predict the LR ($R^2 \geq 0.87$).

**Table 4.** Generic predictive equations for the LR (t ha$^{-1}$) modelled using the three different methods.

| Equation | Generic Predictive Equation | $R^2$ | $n$ |
|:---:|:---:|:---:|:---:|
| (8) | $\mathrm{LR} = e^{-4.98-0.071\times NV+0.67\times \Delta pH+0.57\times pH+0.028\times Clay} \times \mathrm{BD} \times h$ | 0.88 | 542 |
| (9) | $\mathrm{LR} = \Delta pH \times e^{-3.88-0.069\times NV+0.51\times pH+0.025\times Clay} \times \mathrm{BD} \times h$ | 0.87 | 542 |
| (10) | $\mathrm{LR} = \Delta pH/e^{3.93+0.068\times NV-0.51\times pH-0.026\times Clay} \times \mathrm{BD} \times h$ | 0.88 | 542 |

LR is the lime requirement, ΔpH is the difference between soil target pH and soil initial pH, NV is the neutralizing value of lime (mmol kg$^{-1}$), BD is soil bulk density (g cm$^{-3}$), and h is the depth of soil (cm).

Figure 3 shows the relationships between the actual LR and predicted LR estimated via exponential predictive equations. It can be seen that the larger $R^2$ was, the more data were within the 95% prediction bands. Furthermore, Table S6 (see Supplementary Materials for Table S6) shows that although the RMSE was increased with the LR, the minimum MAPE was found in the range of 6~9 t ha$^{-1}$, revealing that the deviations were the least when 6 < LR ≤ 9 t ha$^{-1}$. The ME was positive when the LR was below 6 t ha$^{-1}$ and negative when 6 < LR ≤ 12 t ha$^{-1}$, indicating that the three equations all overestimated the LR when it was below 6 t and underestimated it in the range of 6~12 t ha$^{-1}$. However, over the whole LR range, the MEs were nearly zero, implying that the three predictive equations produced no systematic errors. Equation (8) had the lowest RMSE, MAPE, and absolute value for ME but the maximum $R^2$ value. Therefore, Equation (8) was best at predicting the LR according to the incubation study. Nevertheless, the predictive equation needs to be further validated using field trial data because there are differences between field conditions and the laboratory environment.

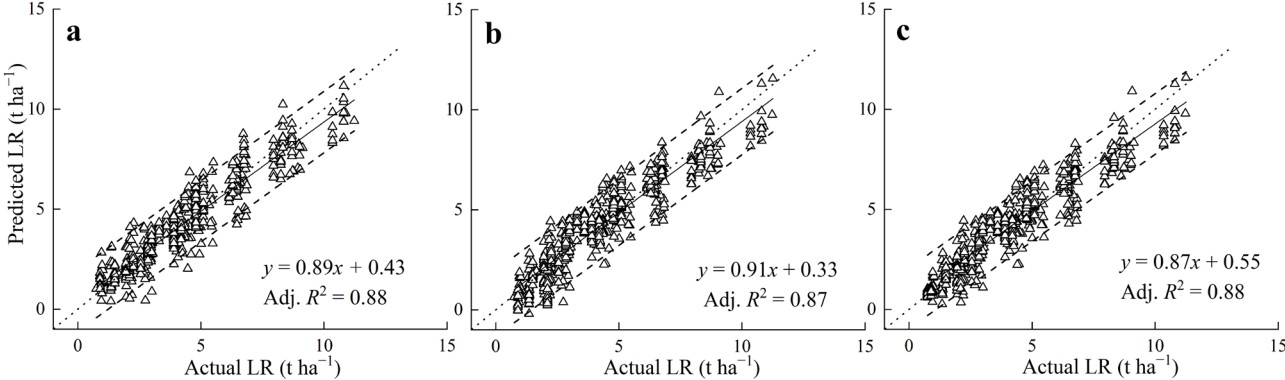

**Figure 3.** Relationships between the predicted LR estimated via Equation (8) (**a**), Equation (9) (**b**), Equation (10) (**c**), and the actual LR. Solid lines denote the linear regressions, dotted lines are the 1:1 lines, and dashed lines are the upper and lower 95% prediction bands.

### 3.4. Model Validation with Field Studies

Figure 4 shows that there are good relationships between the actual LR under field conditions and the predicted LR estimated via Equations (8)–(10) based on the incubation

data ($R^2 \geq 0.78$). The proposed models were validated with field studies, and the results showed that they accurately predicted the LR under field conditions. Moreover, the slopes of the linear relationship between the actual LR and predicted LR were approximately 1.0, indicating that the amount of lime applicated during the incubation experiment was in the same range as that applied in the field studies. Assuming that the soil bulk density was 1.25 g cm$^{-3}$, and there was 1500 t of surface soil per hectare, then it can be deduced that the depth of the surface soil in the fields that were actually neutralized by lime is 12 cm.

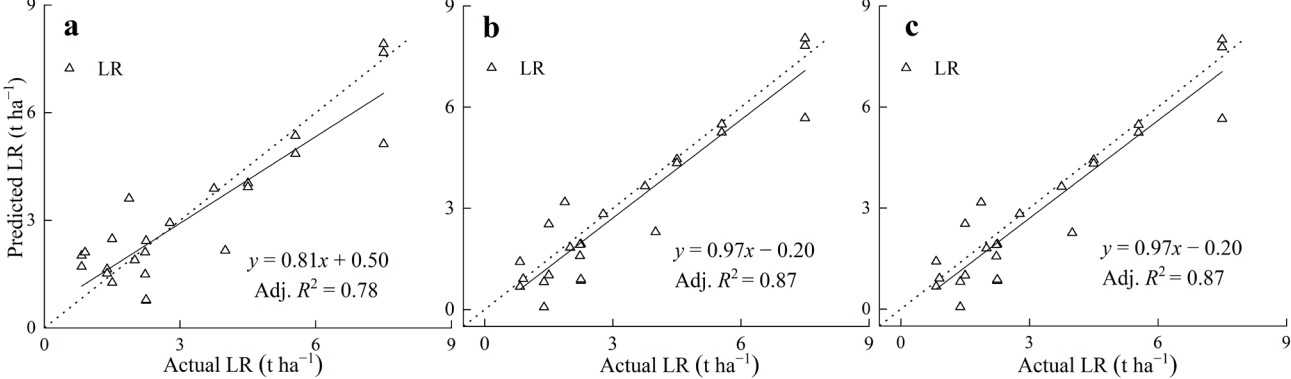

**Figure 4.** Relationships between the predicted LRs predicted via Equation (8) (**a**), Equation (9) (**b**), and Equation (10) (**c**) and the actual LRs derived from field studies. Solid lines denote the linear regressions and dotted lines are the the 1:1 lines.

Based on the $R^2$ result, Equations (9) and (10) could more accurately predict the LR than Equation (8). Furthermore, as shown in Table S7 (see Supplementary Materials for Table S7), the RMSE, MAPE, and ME estimated via Equation (9) were less than that via Equation (10), indicating that the LR deviation was lowest when predicted via Equation (9). Therefore, Equation (9) should be used to predict the LR under field conditions.

The clay content can be estimated using soil texture because soil texture mainly depends on the clay content of the soil. This study assumed that the clay contents in sandy soil, loamy soil, and clayed soil were 5%, 20%, and 45%, respectively, and the depth of soil limed was 20 cm. Furthermore, the soil bulk density can be derived from the relationship between soil bulk density and clay content using the equation y $= 1.52 - 0.00646$x [45]. Quicklime has the greatest neutralizing ability, and limestone is the most commonly available lime that is stable. The proposed generic predictive Equation (9) can be simplified as Equations (11)–(13) for quicklime and Equations (14)–(16) for limestone when predicting the LR for sandy soil, loamy soil, and clayed soil (Table 5). This simplifies the LR calculation and provides more liming options.

**Table 5.** Simplified predictive LR equations (t ha$^{-1}$) for quicklime and limestone.

| Equation | Simplified Predictive Equation | Lime Source | NV (mmol kg$^{-1}$) | Clay (%) | Bulk Density (g cm$^{-3}$) | Soil Texture |
|---|---|---|---|---|---|---|
| (11) | LR $= 0.12 \times \Delta\text{pH} \times e^{0.51 \times \text{pH}}$ | Quicklime | 25 | 5 | 1.49 | Sandy soil |
| (12) | LR $= 0.17 \times \Delta\text{pH} \times e^{0.51 \times \text{pH}}$ | Quicklime | 25 | 20 | 1.39 | Loamy soil |
| (13) | LR $= 0.25 \times \Delta\text{pH} \times e^{0.51 \times \text{pH}}$ | Quicklime | 25 | 40 | 1.26 | Clayed soil |
| (14) | LR $= 0.20 \times \Delta\text{pH} \times e^{0.51 \times \text{pH}}$ | Limestone | 18 | 5 | 1.49 | Sandy soil |
| (15) | LR $= 0.27 \times \Delta\text{pH} \times e^{0.51 \times \text{pH}}$ | Limestone | 18 | 20 | 1.39 | Loamy soil |
| (16) | LR $= 0.41 \times \Delta\text{pH} \times e^{0.51 \times \text{pH}}$ | Limestone | 18 | 40 | 1.26 | Clayed soil |

LR is the lime requirement, and $\Delta$pH is the difference between soil target pH and soil initial pH.

## 4. Discussion

### 4.1. Relationship between NV and [Ca+Mg] in Lime

The results showed a significant correlation between NV and [Ca+Mg], which is probably due to calcium magnesium oxide and calcium magnesium carbonate being the main components in lime that neutralize acid [3,46].

In previous studies, the calcium carbonate equivalent or relative NV, which represents the percentages of $CaCO_3$ or CaO, was usually used for evaluating the neutralizing ability of lime [30,47] or other liming materials, such as byproduct materials [30] and biological ash [48,49]. The liming materials in some studies had a low neutralizing capacity, but only the composition of the lime was reported [50–52]. This study quantified the relationships between lime composition ([Ca+Mg]) and NV. The results from this study suggest that the NV can be estimated using the [Ca+Mg] in lime. Furthermore, the [Ca+Mg] in lime could be also labelled on commercial lime products as an indicator of neutralizing ability. The magnesium contents in quicklime, hydrated lime, and limestone were relatively small [53]. Therefore, the NVs for quicklime, hydrated lime, and limestone can be estimated from the calcium content. However, there is very little magnesium in dolomite [54], so the calcium and magnesium contents were both necessary for NV estimation.

### 4.2. Relationship between the LBC or LR and NV of Lime

The order of average neutralizing ability was quicklime > hydrated lime > limestone > dolomite. Therefore, it can be deduced that the amount of lime required for soil liming was in reverse order. The average NV of limestone is the same as dolomite, but the average LBC of limestone is a little lower than that of dolomite. This result suggests that limestone has a similar neutralizing ability to dolomite in solution but is a little higher than that of dolomite under soil conditions. This may be because the limestone is dissolved more completely than dolomite under soil conditions due to the hardness of limestone being lower than that of dolomite [55].

There was a negative linear relationship between the LBC and NV of lime, indicating that the amount of lime required is less when the NV of lime is high. It can be deduced that there is an approximately negative linear relationship between the LR and NV of lime and that the LR can be calculated from the LBC multiplied by the difference between the target pH and soil pH [17,56].

### 4.3. The Relationship between the LBC or LR and Soil pH, Clay Content

In accordance with previous studies, the LBC or LR was positively correlated with soil clay content [19,22,26], indicating that sandy soil was more sensitive to acid or alkali than loam soil. This finding suggests that soils with low clay contents should be monitored and limed regularly to prevent soil acidification. Furthermore, because of the sharp rise in pH after the addition of lime [47,57], lime should be applied cautiously, especially to soils with a low LBC.

The LBC was positively correlated with soil pH in this study. This was in contrast to previous studies, which reported that the pHBC was negatively correlated with soil pH [26,58]. This difference was probably due to the fact that the lime used in the LBC determination process must dissolve first so that it can then be neutralized by soil acid. The soil pH is an important factor affecting the dissolution of lime in soil [59], and it becomes more difficult to dissolve lime as the soil pH increases. This suggests that there may be remaining undissolved lime in soils with high pHs when incubation is accomplished.

The LBC indicates the ability of soil to neutralize lime [15,17,18], and it can more accurately predict liming under field conditions than pHBC. Therefore, the LBC has an advantage over pHBC when estimating the ability of soil to resist pH change after the addition of lime. Furthermore, the LR was also positively correlated with the soil initial pH. This may be due to the lime dissolving less easily as the soil pH increases [59], which means that more lime is needed for the soil to attain the target pH.

*4.4. The Predictive Equation for the LBC or LR*

The results from this study suggested that it is feasible to estimate the LBC from soil pH, clay, and the NV using an exponential relationship between the LBC and the properties of soil and lime, which was in contrast to the linear relationships between the LBC and soil properties established by previous studies [20,58,60]. This result also indicates that soil pH and clay have a crucial influence on the LBC [19,22,26].

Soil pH has been reported as being just an indicator of the need for liming, not a reliable predictor of the LR [11,21,46]. However, in this present study, pH was a vital factor in LR modelling, indicating that soil pH was not only an indicator of the need for liming but also a reliable predictor of the LR. This difference might be due to the modelling methods. Furthermore, when the predictive equation is applied, the soil pH can be compared to the pH values produced by models that use soil pH determined via other methods [44,61,62].

The source of lime estimated via models in previous studies was restricted to pure agents, for example, pure $CaCO_3$ [10,11], whereas the neutralizing capacity of lime products containing components with a wide range of neutralizing abilities, such as limestone, quicklime, dolomite, and hydrated lime, can be predicted via the exponential model proposed in this study. Moreover, the models produced from some previous studies could only calculate the LR for a target pH of 5.8 or 6.0 [11], and 5.5 or 6.5 [63], whereas the model developed in this study can target a pH that is anything less than 7.50.

Simplified predictive LR equations for different textured soils when neutralized by quicklime and limestone were developed in this present study based on the assumption that the soil initial pH was 5.0 and the clay contents in sandy, loamy, and clay soils were 5%, 20%, and 45%, respectively. The relationship between soil bulk density and clay [45] suggests that the LRs needed for sandy, loamy, and clay soils to change by one unit of pH are 1.2, 2.2, and 3.2 t ha$^{-1}$, respectively, when neutralized by quicklime; the LRs needed are 2.6, 3.5, and 5.3 t ha$^{-1}$, respectively, when neutralized by limestone. The neutralizing ability of hydrated lime is a little lower than that of quicklime, and the neutralizing ability of dolomite is similar to that of limestone. Therefore, the amount of hydrated lime and dolomite required can be referenced from quicklime and limestone. Moreover, the LR proportions from commercially available quicklime, hydrated lime, dolomite, and limestone were 0.58:0.64:0.97:1.00.

## 5. Conclusions

An exponential model integrated with soil pH, clay, and the NV of lime is proposed for predicting the amount of lime product that is required to change the pH of acidic soils to a predetermined target pH in China when the product contains components with a wide range of neutralizing abilities, and the soils have a wide range of soil properties. The results showed that the NV of lime can be derived from the [Ca+Mg] in lime. In addition, the LR proportions for commercially available quicklime, hydrated lime, dolomite, and limestone were 0.58:0.64:0.97:1.00. The proposed model was validated with field studies, and the results showed that it could accurately predict the LR under field conditions using the NV of lime, soil pH, and clay content. The results are of great significance to agronomic activities.

**Supplementary Materials:** The following supporting information can be downloaded at: https://www.mdpi.com/article/10.3390/agronomy13071860/s1, Table S1: Descriptive statistics of properties of soil and lime; Table S2: Selected physical and chemical properties of acidic soil samples; Table S3: Lime source, calcium, and magnesium in lime samples; Table S4: The data used in model validation; Table S5: Descriptive statistics of neutralizing value (mmol kg$^{-1}$) of lime; Table S6: The RMSE, MAPE, and ME between the actual LR and predicted LR estimated via prediction equations; Table S7: The RMSE, MAPE, and ME between the actual LR in field condition and predicted LR estimated by prediction. References [64–70] are cited in the supplementary materials.

**Author Contributions:** Data curation, D.H. and S.Z.; Formal analysis, D.H., X.Z. and S.Z.; Funding acquisition, Y.M. and J.L.; Investigation, X.Z. and S.Z.; Methodology, D.H. and Y.M.; Project administration, J.L.; Supervision, Y.M.; Validation, Y.M.; Visualization, X.Z.; Writing—original draft, D.H.; Writing—review and editing, Y.M. All authors have read and agreed to the published version of the manuscript.

**Funding:** This work was supported by the Science and Technology Development Fund, Macau SAR (0008/2022/AFJ) and the National Key Research and Development Program of China (2016YFD0800406).

**Data Availability Statement:** The data of this study are available from the corresponding author upon reasonable request.

**Conflicts of Interest:** The authors declare no conflict of interest.

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
