# Peer review of "Integrating Soil pH, Clay, and Neutralizing Value of Lime into a New Lime Requirement Model for Acidic Soils in China"

_agronomy, doi:10.3390/agronomy13071860_

Round 1

Reviewer 1 Report

The article “Integrating Soil pH, Clay, and Neutralizing Value of Lime into a New Model of Lime Requirement or Acidic Soils in China” aimed to develop a model that integrated both soil properties (pH and clay) and lime properties (NV) for predicting the required amount of lime, with a wide range of neutralizing abilities, for acidic soil in China with a wide range of soil properties to attain desired soil pH. The article is well written and can be published after some revisions.

For example. The English language of the MS should be improved through some native English speaker or some professional facility, this is evident from the first sentence of the abstract.

Table 1 donot provides any significant information, such tables should be sent to supplementary and info like median values and st dev are not needed to be interpreted.

Same is the case with table 3

The discussion is very poorly written with no support of citations and explanations in many sections, that mostly looks repeated with results. For example Section 4.2 and 4.4. Both of these sections need major revisions

moderate editing required

Author Response

Dear reviewer:

Thank you for the time and effort for reviewing the previous version of the manuscript. Your suggestions have enabled us to improve our work. We have carefully revised the manuscript and made some changes. These changes will not influence the content and framework of the paper. Meanwhile, the language of the manuscript has been thoroughly revised and polished by a native English editor.

we have uploaded a copy of the original manuscript with the major changes highlighted in yellow. The following is our point-by-point response to the comments. The comments are reproduced and our responses are given directly afterward in red.

Thanks again!

Reviewer 2 Report

The manuscript is about determining the lime requirement for Chinese agricultural soils. Lime is required in non-calcareous agricultural soils to neutralise the effect of various fertilisers, acid deposition, and leaching of HCO3. Many developed countries have specific advices for their soils and crops which are already rather old and which have rarely been published in peer reviewed papers. It is therefore relevant to encourage publication of manuscripts on this subject.

The authors hardly cite or use any work from other researchers, and do not cite the huge amount of work that has been put in international standardisation by ISO (world) and CEN (Europe)(see CEN/TC 260) or methods in the US (AOAC). Their equation cannot be used by other people because the use other methods for soil pH, neutrilising value, and pretreatment of the lime materials.  

-determining the lime requirement of soil, flowing various methods

-determining the neutralizing value, titrimetric methods. Please note the exceptions for silicate and P containing liming materials. (ISO 20978)

-determination of reactivity. This is relevant because not all liming materials work quickly, due to large particles, or due to their chemical contents. 

The authors neglect that many silicate materials (clean wastes from industry) are used for liming. In my county 50% of all limes used in agriculture are “waste” materials, so it is rather relevant.

The authors do not use the international used unit for neutralizing value: eq CaO or eq CaCO3.

The authors do not use a soil pH determination that is common for soil laboratories: most use extracts of KCl or CaCl2 (with good reasons). So how to use you equation?

I think the work should be rewritten. The authors should note that the methods are well-known, and what they do is just a validation for their region. Also the experimental section is not complete: somebody else should be able to repeat the work.  

remarks

66. “there is no model”. That is a rather incorrect statement. There are various models, and apps (https://aglime.org.uk/lime_calculator.php: noted by the authors in line 124) etc available for farmers depending on the country. Maybe you refer only to China. Please be clear, and make reference to abundant literature outside China.

For tropical soil: https://www.ncbi.nlm.nih.gov/pmc/articles/PMC10033874/

In Brazil: https://www.scielo.br/j/rbcs/a/3jtxgqxSF8VYbdnjd6jSTPg/?lang=en

In California: https://www.cdfa.ca.gov/is/ffldrs/frep/pdfs/completedprojects/01-0511miller2005.pdf

76 “representative acidic soil .. from China”. How? Be clear. If somebody tries to repeat this, and ends up with different soil, it should be clear how the method differ. It is not bad if you just guessed it on the basis of a soil map, but please make such statements. How did you select the sampling locations? Are these test locations, or sites from farmers, or locations of which you had previous measurements? Please be clear about the selection method.

81-90. The methods are given with a very short description. Have local protocols been used, have international protocols been used, and have accredited labs been used? Have controls been used? Nobody can be repeat or check your work if you write in this manner.

126 “when the incubation was accomplished”. This rather vague. Please mention the time: after 1 year or two years. If the pH is rather high, not all lime will have reacted.  You mention this yourself in line 342.

134 “three duplicates”: duplo means two. You mean three replica’s. Please get help form an English speaking person.

391 and 398. These simplifications that work for your samples. You cannot conclude that these always work.

Table S1. The authors have used 17 soils and various lime samples to derive the lime requirement on the basis of the target pH and initial pH and soil characteristics.  The soils seems to have rather low organic matter contents, high amounts of clay and silt, and are therefore rather specific. The sampling sites are not given. It is common to give XY coordinates to help future research.

Table S2. The chemical analysis is not provided, except Ca and Mg. It is well known that Fe and S content in lime can influence the neutralising value. The neutralizing effect is also influence by the particle size. Have the particle sizes not been determined?

see above

Author Response

(The authors gave the same response as above.)

Round 2

Reviewer 2 Report

The authors have made appropriate changes. I regret they have not used international methods for determining pH, neutralizing value and reactivity, and I also regret that they not mention this.